# Lying in a 3T MRI scanner induces neglect-like spatial attention bias

**Axel Lindner[1,2]\*[†], Daniel Wiesen[1][†], Hans-Otto Karnath[1,3]\*[†]**

[1]Centre of Neurology, Division of Neuropsychology, Hertie-Institute for Clinical Brain Research, University of Tübingen, Tübingen, Germany; [2]Tübingen Center for Mental Health (TüCMH), Department of Psychiatry and Psychotherapy, University of Tübingen, Tübingen, Germany; [3]Department of Psychology, University of South Carolina, Columbia, United States

**Summary** The static magnetic field of MRI scanners can induce a magneto-hydrodynamic stimulation of the vestibular organ (MVS). In common fMRI settings, this MVS effect leads to a vestibular ocular reflex (VOR). We asked whether – beyond inducing a VOR – putting a healthy subject in a 3T MRI scanner would also alter goal-directed spatial behavior, as is known from other types of vestibular stimulation. We investigated 17 healthy volunteers, all of which exhibited a rightward VOR inside the MRI-scanner as compared to outside-MRI conditions. More importantly, when probing the distribution of overt spatial attention inside the MRI using a visual search task, subjects scanned a region of space that was significantly shifted toward the right. An additional estimate of subjective straight-ahead orientation likewise exhibited a rightward shift. Hence, putting subjects in a 3T MRI-scanner elicits MVS-induced horizontal biases of spatial orienting and exploration, which closely mimic that of stroke patients with spatial neglect.

**\*For correspondence:**
a.lindner@medizin.uni-tuebingen.de (AL);
karnath@uni-tuebingen.de (H-OK)

[†]These authors contributed equally to this work

**Competing interest:** The authors declare that no competing interests exist.

## Introduction

*Roberts et al., 2011* observed that healthy subjects who were exposed to the static magnetic field of 3T and 7T MRI scanners developed a persistent horizontal nystagmus in complete darkness. Since then, this effect was replicated by various groups (e.g. *Mian et al., 2013*; *Boegle et al., 2016*) and was shown to be present for standard fMRI settings at 1.5T and 3T (*Boegle et al., 2016*). Moreover, these studies have further corroborated the initial conclusion by *Roberts et al., 2011* that the nystagmus is caused by Lorentz forces that result from the interaction of the static magnetic field of the scanner and ionic currents in the endolymph fluid of the subject's labyrinth (see *Ward et al., 2019* for review). This force acts upon the cupulae of the horizontal and anterior semicircular canals, inducing a mixed horizontal and torsional nystagmus (*Otero-Millan et al., 2017*). The stronger horizontal component comprises of a slow horizontal vestibular ocular reflex (VOR) that is accompanied by fast resetting saccades in the opposite direction. In natural settings, a stimulation of the horizontal canals occurs, for instance, when our head is accelerated to the left or to the right. The resulting VOR then smoothly moves our eyes in the direction opposite to the head rotation. The VOR thus supports – together with the visually induced optokinetic reflex – the stabilization of the retinal image despite head-motion (for review e.g. see *Ilg, 1997*; *Angelaki and Cullen, 2008*).

*Roberts et al., 2011* (p.1638) pointed out that the fake head-motion signal induced by magneto-hydrodynamic vestibular stimulation (MVS) 'carries important ramifications and caveats for functional MRI studies, not only of the vestibular system but of cognition, motor control, and perception in general'. For instance, *Boegle et al., 2016* observed that MVS does indeed modulate fMRI-signal fluctuations. This modulation affected the default mode network as a function of the strength of the magnetic field and mainly in areas associated with vestibular and oculomotor function. Thus, MVS

induces balance shifts in resting-state network dynamics that might be 'like a "special patient group with a vestibular imbalance" but without lesions in the inner ear or central nervous system' (*Boegle et al., 2016*; p.420).

However, do these functional modulations by MVS also alter behavioral responses? Would the mere exposure to the uniform magnetic field within a 3T MRI scanner already produce the characteristic biases of spatial orientation and exploration that we know to occur with vestibular stimulation? For example, in healthy subjects, caloric vestibular stimulation (CVS) of one external auditory canal not only induces a VOR but also provokes 'neglect-like' behavioral phenomena. Stroke patients with spatial neglect show a tonic deviation of head and eyes towards the ipsilesional side and no longer explore or orient towards large parts of space contralateral to their lesion and therefore neglect contralesionally located people or objects (*Heilman et al., 1983*; *Karnath, 2015*, for a review). In healthy volunteers, CVS induces a tonic bias of head orientation around the yaw axis (*Karnath et al., 2003*) and a tonic shift of the average horizontal eye position (*Abderhalden, 1926*; *Jung, 1953*). Both of these orientation biases resemble those observed in neurological patients suffering from spatial neglect (cf. *Fruhmann-Berger and Karnath, 2005*): already at rest, that is when doing nothing, these patients' head and eyes are tonically deviated towards the ipsilesional side. Moreover, CVS in darkness even induces a bias in subjective straight-ahead (SSA) in healthy subjects, mimicking the bias of the SSA in neglect patients towards their ipsilesional side (e.g. *Karnath et al., 1994c*; *Chokron and Imbert, 1995*; *Kapoor et al., 2001*). Finally, CVS biases healthy subjects' scan path during visual search (*Karnath et al., 1996*): when exploring their surroundings for possible targets subjects' eye movements are no longer symmetrically distributed in the horizontal dimension but biased towards one side of the body's midsaggital plane. Also this CVS-induced horizontal bias in spatial attention is resembled by the spontaneous, asymmetrical, spatially biased exploratory behavior of neglect patients (*Karnath et al., 1996*; *Karnath et al., 1998*), leading to 'spatial neglect' of one side of the surrounding scene. Conversely, it is possible to compensate the spontaneous biases of neglect patients through CVS (*Rubens, 1985*; *Karnath, 1994b*; *Vallar et al., 1995*; *Karnath et al., 1996*; *Rode et al., 1998*; for review e.g. see *Rossetti and Rode, 2002*).

In the present study, we asked whether these behavioral effects on spatial orientation and exploration known from CVS are likewise induced by MVS through exposure to a static magnetic field of a 3T MRI scanner. We hypothesized that MVS should induce a horizontal bias in both the exploratory scan path during visual search and in the SSA of healthy subjects, resembling the horizontal biases of neglect patients.

## Results

To address our research question we analyzed oculomotor behavior of 17 healthy subjects inside and outside a standard 3T MRI scanner (Siemens MAGNETOM Prisma). All subjects provided their informed consent according to the guidelines of our institutional ethics board.

Parts of our procedures resembled that of typical fMRI experiments (compare *Figure 1—figure supplement 1* and Materials and methods for further details): We placed our subjects with their back on the scanner table and their head rested inside a 20-channel head-coil. A mirror was mounted on top of the head-coil to provide subjects with an indirect view on our custom-made black 'visual search-screen'. The screen was placed behind the coil inside the scanner bore at about 110 cm viewing distance. Various glass-fiber-cables penetrated the screen at discrete locations and allowed us presenting small visual stimuli by feeding these cables with LED light signals. Maximal target distance from the screen center amounted to ±12° visual angle in the horizontal direction and to –5° to 6° in the vertical direction. An IR-video-camera unit was mounted next to the mirror to record a video-signal of subjects' right eye. The study was conducted in complete darkness by extinguishing all light sources and covering subjects with a black blanket. This was important, as any visible stimulus that would be present in subjects' visual field, could help them to suppress a MVS-induced VOR as well as to explore the visual scene and to perform the SSA task in an unbiased fashion.

To probe for the putative MVS-effects on spatial orienting and exploration we designed an experiment comprising of three consecutive phases: (i) an initial '*outside 1*' phase with our subjects on the scanner-table at its maximal horizontal displacement from the scanner center (head coil at ~125 cm horizontal [0 cm vertical] displacement); the strength of the magnetic field in this position is roughly 10 times smaller than at the scanner center (compare *Figure 1—figure supplement 1*); (ii) a subsequent

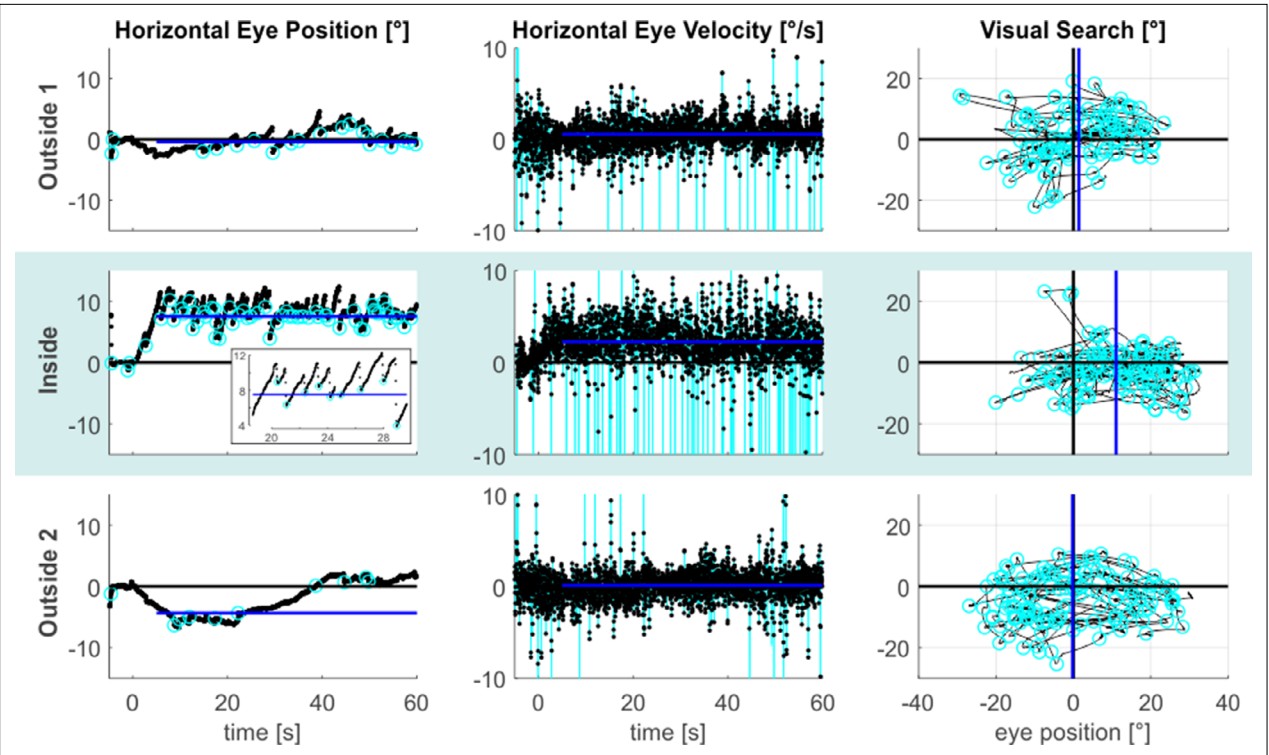

**Figure 1.** MVS-induced oculomotor behavior. The figure provides exemplary data from an individual subject. The three task phases are represented by individual rows. Horizontal eye position data from the looking straight-ahead task are shown in the left column. The figure inset additionally expands parts of the *inside* time course to better illustrate the alternation between the slow phase VOR toward the right and the fast resetting saccades directed in the opposite direction. Saccade endpoints are depicted by the cyan circles and the blue lines reflect the respective SSA estimates. Corresponding horizontal eye velocity traces of the looking straight ahead-task are shown in the middle column. Note that the cyan peaks in these time-courses refer to individual saccades, which have been removed from the eye velocity records to allow estimation of slow-phase velocity in isolation (the blue lines indicate the respective horizontal VOR-estimates). Finally, 2D eye-position eye data from the visual search task are depicted in the rightmost column (time periods with search targets plus 5 s are excluded). The horizontal center of visual search is depicted by the blue vertical lines, reflecting the median of horizontal saccade endpoints (cyan circles). Positive values indicate the rightward/upward direction.

The online version of this article includes the following figure supplement(s) for figure 1:

**Figure supplement 1.** Experimental setup.

**Figure supplement 2.** 'Feet first' control experiment.

'*inside*' phase with the subject's head being in the center of the scanner bore where the strength of the uniform magnetic field is 3 Tesla; (iii) as well as a final '*outside 2*' phase that was identical to *outside 1*. Accordingly, any MVS effect on behavior or attention that we would observe during these three phases should be stronger for *inside* as compared to the two *outside* phases. Each phase started with an initial '*calibration task*' for eye-tracking (see Materials and methods section for details). We next instructed subjects to fixate at a dim light-point presented in the center of their visual field for 5 s. Then the dim light-point was switched off and the subjects had to 'look straight ahead' for 60 s in complete darkness ('*looking straight-ahead task*'). Finally, we asked subjects to find and fixate transient light stimuli. During this '*visual search task*', we presented six search targets (each slowly fading-in over a period of 5 s). Locations of search targets were pseudorandomized across phases and between participants. The presentation of these stimuli only served to maintain the subjects' motivation to search for possible targets. Our interest was the spatial distribution of subjects' overt spatial attention, as assessed by their exploratory scan path in the absence of any visual targets. Thus, for most of the time (i.e. 140 s; total duration of visual search task: 172 s), no visible target was present and subjects were searching in complete darkness. For data analysis we discarded the time periods with targets present (plus an additional grace period of 5 s after a light stimulus was turned off; this was done to prevent carry-over effects from prior target-fixation). The search task always ended with

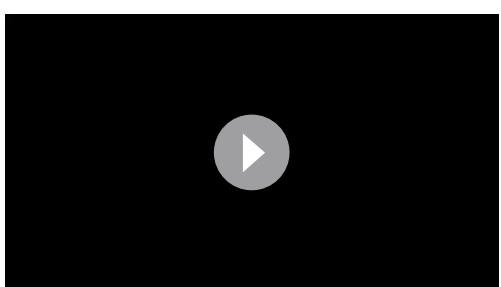

**Video 1.** This video was taken after completion of our experiment. It serves to illustrate the MVS-induced effects on eye movements in complete darkness. There was no obvious horizontal nystagmus when the subject was outside the scanner (at a position identical to o*utside 1* and *2* phases). A pronounced horizontal nystagmus was present inside the scanner (and already when the subject was being moved into the scanner). The horizontal nystagmus was alleviated when moving the subject outside the scanner. At the outside position virtually no nystagmus was present. Note that longer exposure times to the magnetic field inside the MRI-scanner would typically lead to an aftereffect of the VOR in the opposite direction (e.g. compare **Roberts et al., 2011**).

https://elifesciences.org/articles/71076/figures#video1

a central light stimulus presented during the last 2 s (also not considered for data analysis). Please refer to Materials and methods for detailed information on the time-course of our experiment.

*Figure 1* illustrates the resulting eye-data of an exemplary subject throughout all tasks and phases. It shows that during the *inside* phase the typical saw-tooth-like eye movement pattern, which is characteristic for a vestibular nystagmus, was observed (also compare *Video 1*). It consisted of a slow rightward VOR that was accompanied by fast resetting saccades in the opposite leftward direction. This prototypical nystagmus-pattern was largely reduced for the *outside 1* phase and it was practically absent for the *outside 2* phase. To express the strength of the VOR quantitatively, we calculated the median de-saccaded horizontal eye-velocity (starting 5 s after the offset of the central light stimulus). This estimate is reflected by the blue lines in the respective horizontal eye-velocity time-courses shown in the middle column of *Figure 1*.

The same qualitative effects were present in all of our 17 subjects, as is shown in *Figure 2*. The average slow eye velocity for the *outside 1* phase was 0.30°/s on average ( ± 0.49°/s standard deviation [SD]) and increased to 1.42°/s ( ± 1.49°/s SD) for the *inside* phase. After removing the subjects from the scanner bore to *outside 2*, average slow eye velocity decreased to –0.01°/s ( ± 0.40°/s SD). The within-subject differences in velocity between *outside 1* and *inside* and between *inside* and *outside 2* were significant (one-tailed paired t-test: $t(16)=6.62$, $p < 0.001$, $g_1[CI_{95\%}] = 1.60$ [0.87, 2.32] and $t(16)=6.69$, $p < 0.001$, $g_1[CI_{95\%}] = 1.62$ [0.88, 2.35], respectively). Average horizontal eye velocity during *outside 1* was also slightly larger as compared to *outside 2* (two-tailed paired t-test: $t(16)=3.08$, $p = 0.007$, $g_1[CI_{95\%}] = 0.75$ [0.20, 1.28]). As expected (**Otero-Millan et al., 2017**), the MVS effects affected only horizontal eye movements of our 2D eye movement recordings and were absent in the vertical direction (*outside 1*: –0.29°/s ± 0.56°/s SD; *inside*: –0.29°/s ± 0.60°/s SD; *outside 2*: –0.21°/s ± 0.58°/s SD; statistical comparisons with two-tailed paired t-tests yielded no significant differences in vertical eye velocity between any of the three task phases [$p > 0.20$]). In summary, these data agree with earlier observations of an MVS-induced horizontal VOR (for review see **Ward et al., 2019**) and provide additional evidence for the consistent presence of this effect while being inside a 3T MRI scanner (**Boegle et al., 2016**).

We next asked whether MVS also affects the distribution of overt attention in our visual search task. The rightmost column in *Figure 1* demonstrates a 2D-plot of our exemplary subject's eye position during search, including only those time epochs when no target stimulus was presented. While the area of visual search for both *outside* phases was centered on the middle of the screen (which was aligned to the subject's body midline), the search space clearly deviated to the right when the subject was *inside*, while parts on the far left of the search-space were now neglected. This horizontal bias in our subject's exploratory eye movements was not a mere consequence of the rightward VOR inside the scanner but was likewise reflected by the distribution of overt attention, that is the endpoints of the saccades guiding visual search: the mean horizontal position of these saccade endpoints (indicated by the blue vertical lines in the rightmost plots in *Figure 1*) almost perfectly resembled the center of the search screen for the *outside 1 and 2* phases, while it was clearly shifted to the right for the *inside* phase. Across our 17 subjects, statistical analysis of subjects' exploratory spatial behavior (*Figure 3*) revealed that the average horizontal mean of visual search was at 1.65° ( ± 4.40° SD) and at 1.01° ( ± 3.98° SD) for the *outside 1* and *2* phases, respectively, while *inside* this measure was significantly shifted to 4.69° ( ± 4.61°) towards the right (one-tailed paired t-tests; *outside 1* vs. *inside*:

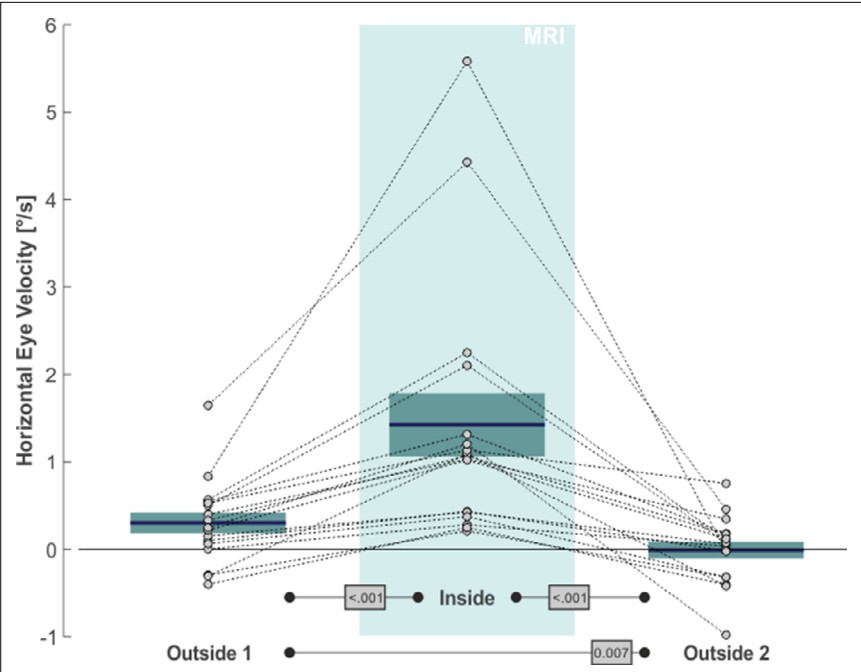

**Figure 2.** MVS-induced increase in de-saccaded horizontal eye velocity towards the right. Horizontal eye velocity of individual subjects (circles, N = 17) are shown for all three task phases. Positive values denote rightward motion. The blue lines and the boxes reflect across subjects mean and standard error (SE; to account for our within-subject design, we removed the between-subject variance for calculation of the SE according to the procedure described by *Masson and Loftus, 2003*). The p-values of pairwise statistical comparisons are indicated in grey boxes.

The online version of this article includes the following figure supplement(s) for figure 2:

**Figure supplement 1.** Increase in de-saccaded horizontal eye velocity towards the left during 'feet first' control experiment.

**Figure supplement 2.** MVS-induced changes in VOR predict spatial biases in visual search and SSA.

t(16)=-5.12, p < 0.001, $g_1[CI_{95\%}]$ = 1.24 [0.59, 1.87]; *inside* vs. *outside 2*: t(16)=4.38, p < 0.001, $g_1[CI_{95\%}]$ = 1.06 [0.45, 1.65]). The measures did not significantly differ between phases *outside 1* and *2* (two-tailed paired t-test: t(16)=0.76, p = 0.46, $g_1[CI_{95\%}]$ = −0.18 [-0.66, 0.30]).

Finally, we asked whether the MVS-induced spatial bias would also affect the looking straight ahead task. To this end, we again focused on the horizontal distribution of saccade endpoints during this task, as they reflected subjects' voluntary attempt to continue looking straight ahead. For each subject and for each task phase, we calculated the median horizontal position of saccade endpoints (starting 5 s after the offset of the central light stimulus) as a proxy for the subjective straight-ahead (SSA). This measure is illustrated by the blue lines in the leftward column of *Figure 1* for our exemplary subject. Similar to the horizontal shift of exploratory search behavior across task phases, also the looking straight ahead measure shifted. Initially, the SSA was close to the midline during the *outside 1* phase. It shifted to the right for the *inside* phase and then back towards the left (with some overshoot) for the *outside 2* phase. Across our 17 subjects, the SSA likewise exhibited a significant rightward shift from the *outside 1* (1.79° ± 5.39° SD) to the *inside* (4.76° ± 4.02° SD) phase (one-tailed paired t-test: t(16)=-2.38, p = 0.015, $g_1[CI_{95\%}]$ = 0.58 [0.05, 1.08]; compare *Figure 4*). This effect vanished after removing subjects from the scanner in the *outside 2* phase (0.26° ± 4.37° SD; one-tailed paired t-test: t(16)=3.56, p = 0.001, $g_1[CI_{95\%}]$ = 0.86 [0.29, 1.42]). There was no significant difference between the SSA for phases *outside 1* and *2* (two-tailed paired t-test: t(16)=1.17, p = 0.26, $g_1[CI_{95\%}]$ = −0.28 [-0.76, 0.21]).

One may wonder whether being inside an MRI scanner might have led to a diminution in subjects' alertness and that this – rather than the scanner's static magnetic field – could be the cause of the reported rightward biases in spatial attention (cf. *Manly et al., 2005*; *Chandrakumar et al., 2019* for a meta-analysis) and the perception of SSA. To rule out such possible influence

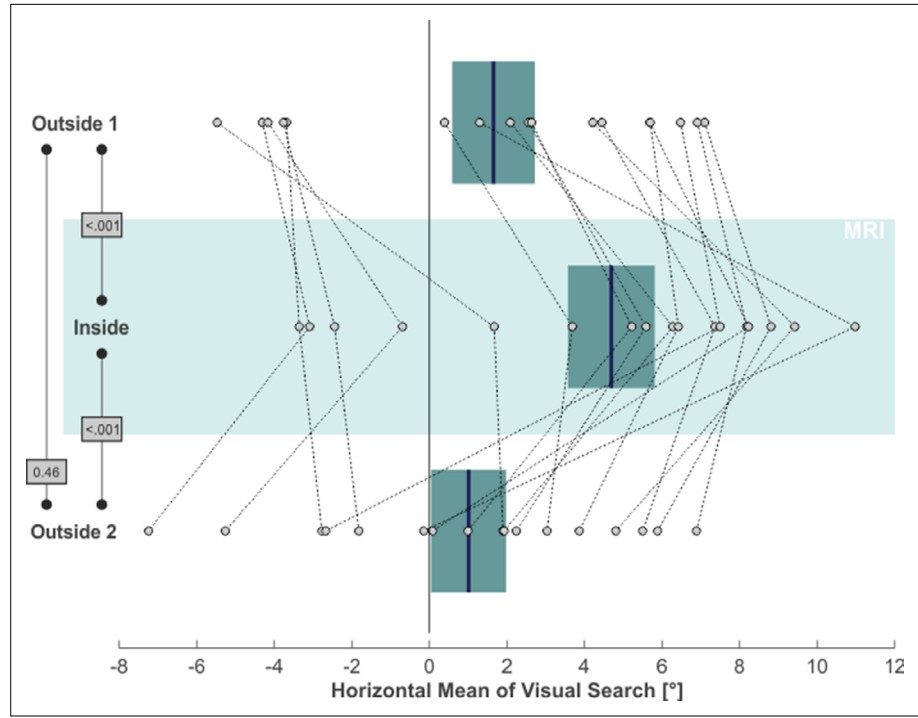

**Figure 3.** MVS-induced rightward bias in visual search. Individual subjects' horizontal mean saccade endpoints during visual search (in the absence of targets) are depicted as individual circles and for all three task phases (N = 17; positive values = right). The blue lines and the boxes reflect across subjects mean and SE (to account for our within-subject design, we removed the between-subject variance for calculation of the SE according to the procedure described by *Masson and Loftus, 2003*). The p-values of pairwise statistical comparisons are indicated in grey boxes.

The online version of this article includes the following figure supplement(s) for figure 3:

**Figure supplement 1.** Leftward bias in visual search during 'feet first' control experiment.

of alertness as well as any other effect on subjects' behavior that could result from asymmetrical cortical representations and/or activation related to arousal, we repeated the measurements of our main experiment in  subjects, who, however, now entered the scanner in the reverse position, that is with their feet first. The rationale behind this manipulation was that the MVS-effect is inverted when subject-positioning is reversed with respect to the static magnetic field vector of the MRI-scanner (e.g. compare *Roberts et al., 2011*), while there is no change in alertness or arousal as compared to our main experiment. In correspondence to the now inverted MVS-effect we expected that the behavioral effects are inverted as well, that is show a bias to the left instead of to the right side. We provide a detailed description of this '*feet first' control experiment* in *Figure 1—figure supplement 2*. In brief, subjects now exhibited a strong leftward VOR *inside* the scanner (–1.48°/s ± 1.43°/s SD) as compared to both *outside 1* and 2 phases (0.36°/s ± 0.30°/s SD & 0.49°/s ± 0.24°/s SD) (see *Figure 2—figure supplement 1*). Importantly, the biases in overt spatial attention (*outside 1*: 1.12° ± 3.39° SD; *inside*: –2.82° ± 4.57° SD; *outside 2*: 2.06° ± 2.05° SD; see *Figure 3—figure supplement 1*) and the perception of SSA (*outside 1*: 2.74° ± 2.76° SD; *inside*: –4.97° ± 3.49° SD; *outside 2*: 1.70° ± 1.46° SD; see *Figure 4—figure supplement 1*) reversed as well. In other words, the spatial biases were now toward the left side and thus in the direction predicted by MVS.

Finally, we performed exploratory regressions to analyze whether the MVS-induced changes in VOR would predict the observed changes in visual search and SSA across our experiments. Indeed, the average change in VOR velocity for the *inside phase* as compared to both *outside 1 and 2 phases* reliably predicted respective changes in both visual search ($R^2 = 0.53$; $F(1,21)=23.43$; $p < 0.001$) and in subjects' SSA ($R^2 = 0.46$; $F(1,21)=18.04$; $p < 0.001$) (also compare *Figure 2—figure supplement 2*).

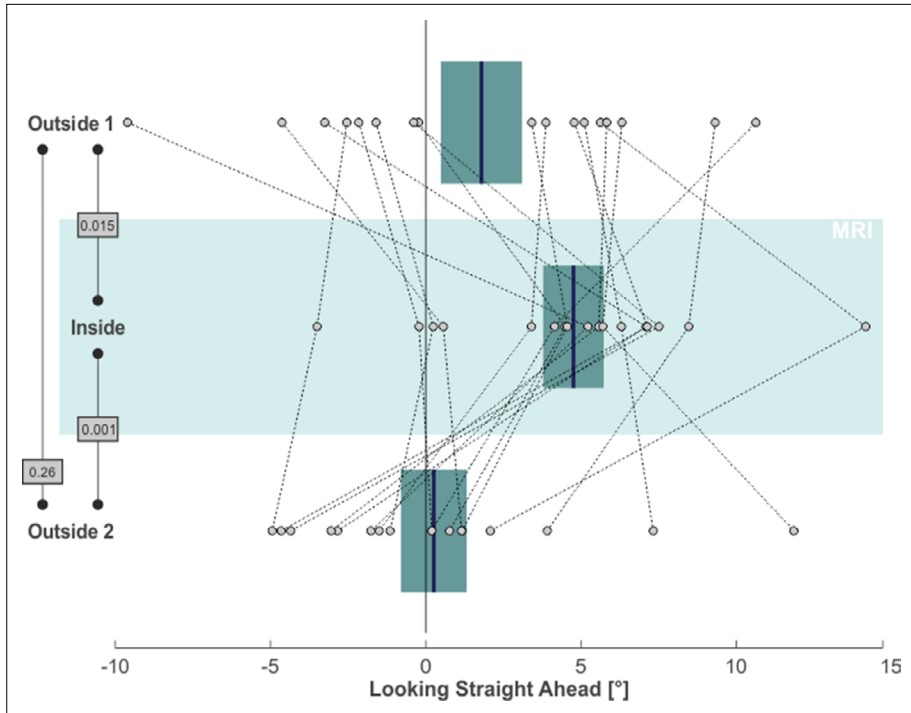

**Figure 4.** MVS-induced rightward bias in subjective straight-ahead (SSA). Individual subjects' SSA, namely the median horizontal saccade endpoint when trying to look straight ahead, is depicted by individual circles for all three task phases (N = 17; positive values = right). The blue lines and the boxes reflect across subjects mean and SE (to account for our within-subject design, we removed the between-subject variance for calculation of the SE according to the procedure described by *Masson and Loftus, 2003*). The p-values of pairwise statistical comparisons are indicated in grey boxes.

The online version of this article includes the following figure supplement(s) for figure 4:

**Figure supplement 1.** Leftward bias in subjective straight ahead (SSA) during 'feet first' control experiment.

## Discussion

Like previous studies (for review see *Ward et al., 2019*), we observed a MVS-induced tonic direction-specific horizontal VOR when lying inside a 3T MRI scanner. As compared to the *outside 1* and 2 phases, this effect was present in every single subject. The velocity of the VOR inside the MRT amounted to 1.42°/s during our looking straight-ahead task. Our task instruction to 'fixate an imaginary target', namely into the direction of a previously illuminated target LED in the medial plane, probably reduced VOR-velocity by about 25 % as compared to a situation without such imaginary fixation (cf. *Schmäl et al., 2000*; *Barr et al., 1976*). Given this latter consideration, our estimate well corresponds to the roughly 2°/s VOR reported in an earlier MVS study, which used a comparable experimental setting at 3T but without imaginary fixation (*Boegle et al., 2016*).

Our experiment further revealed that lying inside the 3T MRI scanner (in the common position in neurology/neuroscience, i.e. with head first) also led to a spatially biased neglect-like behavior when subjects were trying to look straight ahead and when they were performing visual search. First of all, we would like to point out that the respective biases in the looking straight ahead task and the visual search task were not a mere reflection of the tonic VOR-induced drift of subjects' eyes toward the right. The critical behavioral measure in both tasks refers to the distribution of endpoints of subjects' voluntary saccades while they were trying to follow task instructions. The passive drift of subjects' eyes due to the MVS-induced VOR does not influence this measure of active spatial behavior (for further discussion also cf. *Karnath et al., 1996*, p.340). Hence, the reported shifts inside the scanner reflect a true spatial bias in overt spatial attention and active goal-directed behavior: (i) in the visual search task the center of subjects' visual scan-path (in the absence of any visible target) was shifted by roughly 3.4° to the right; (ii) in the looking straight ahead task, the average endpoint of subjects' saccades in darkness likewise deviated toward the right, namely by about 3.7°.

The MVS-induced changes in VOR that we observed in our study did reliably predict the changes in visual search and in the SSA across experiments (compare *Figure 2—figure supplement 2*). Interestingly, we also revealed a significant decrease of the residual rightward VOR from the *outside 1 phase* to the *outside 2* phase. Similarly, also the center of visual search as well as the SSA did numerically differ between *outside 1* and *2* phases, while being closer to midline for the *outside 2* phase. Yet, these latter changes were not significant. The decrease in VOR from the *outside 1 phase* to the *outside 2* phase could be the result of a VOR set-point adaptation that has been shown to occur for longer exposure to MVS at 7T: in the presence of conflicting afferent inputs signaling spatial stability (such as from the otoliths etc.), the nervous system counteracts the MVS-induced tonic imbalance, as is obvious from a slow but incomplete reduction of the VOR over the course of minutes to hours (*Jareonsettasin et al., 2016*). Whether or not the reported biases in the center of visual search and in the SSA could likewise be subjects to set-point adaptation is an interesting question for future research: Our task design was optimized to exhibit qualitative effects of MVS on these measures (compare methods) and does not permit investigating their time-course of adaptation.

All aforementioned results well resemble the known effects of caloric vestibular stimulation on spatial exploration and orientation: After CVS, healthy subjects' estimates about SSA, quantified by verbally instructing orientation of a laser-spot into this direction, shifted horizontally (e.g. *Karnath et al., 1994c*; *Karnath et al., 1996*). Moreover, CVS likewise leads to a horizontal shift of healthy subjects' scan path during visual search (*Karnath et al., 1996*). These known CVS-induced deviations ranged roughly between 4° and 9° and thus were slightly larger than the MVS-induced spatial bias observed in the present experiment. Importantly, the CVS-induced biases in spatial behavior also surface when trying to manually point straight ahead (*Schmäl et al., 2000*) and during manual spatial exploration (*Karnath et al., 2003*), respectively. This suggests that CVS – and probably also MVS – does induce changes in spatial representations for goal-directed actions that are effector-independent. These CVS/MVS-induced alterations of spatial behavior are most likely of cerebro-cortical origin. Notably, there exists no 'primary vestibular cortex' in the strict sense but rather an interconnected multimodal 'cortico-vestibular system' with the parieto-insular vestibular cortex (PIVC) at its core (e.g. see *Guldin and Grüsser, 1998*; *Karnath and Dieterich, 2006*; *Lopez and Blanke, 2011* for reviews). The areas that make up this system further entail neighboring parts in the temporo-parietal junction, posterior parietal cortex, superior temporal cortex, as well as regions in the anterior insula, retroinsular regions, somatosensory cortex, cingulate cortex, premotor cortex, and in the hippocampus (*Guldin and Grüsser, 1998*; *Kahane et al., 2003*; *Karnath and Dieterich, 2006*; *Lopez et al., 2012*; *zu Eulenburg et al., 2012*; *Mazzola et al., 2014*; *Frank and Greenlee, 2018*). Neurons in part of these areas have been shown to integrate vestibular cues (along with neck proprioception and information about eye position) in a way that allows them to represent visual action goals in reference to the body (e.g. *Andersen et al., 1999*; *Cohen and Andersen, 2002*; *Chen et al., 2018*). This body-centered, egocentric representation of spatial information underlies, for example, the guidance of our goal-directed behavior including saccades, reaching, or heading etc. (see *Karnath, 2001* for review). Such 'body referencing' has also been reported in human functional imaging studies (e.g. *Bottini et al., 2001*; *Galati et al., 2010*; *Schindler and Bartels, 2013*; *Saj et al., 2020*). It thus is conceivable that tonic MVS (as well as CVS, etc.) introduces systematic biases in these spatial action-representations, leading to the reported alterations of subjective straight ahead and spatial exploration (also compare *Laurens and Angelaki, 2018*). Similarly, altered body referencing was suggested to explain the horizontal shifts in spatial behavior present in spatial neglect (*Karnath, 1994a*; see *Karnath, 2015* for a review). Moreover, the pattern of lesions exhibited by patients with spatial neglect is largely overlapping with the multimodal cortico-vestibular system and exhibits a common lateralization (*Karnath and Dieterich, 2006*; *Karnath and Rorden, 2012*; *Dieterich and Brandt, 2015*). Hence, MVS induces neglect-like alterations of spatial behavior in healthy subjects and these alterations are supposedly due to its interference with the cortico-vestibular system, the same system that is typically affected in patients suffering from spatial neglect.

Given the aforesaid, the question arises whether MVS could help to ameliorate spatial neglect and thus might become a novel option for treatment. In this context, the observations that healthy subjects exposed to the static magnetic field of an MR scanner developed a persistent horizontal nystagmus that slowly diminished but did not extinguish (*Roberts et al., 2011*; *Jareonsettasin et al., 2016*), is highly interesting since the effect of, for example, CVS only lasts for several minutes. If its

effects on spatial exploration and orientation would follow a similar time-course, the tonic nature of MVS could serve as a noninvasive and comfortable means to continuously stimulate the labyrinth of neurological patients with spatial neglect and might help to induce longer-term plastic changes in patients' pathologically altered spatial representations (cf. *Jareonsettasin et al., 2016*). Against this possible advantage over CVS and over other types of vestibular stimulation (*Kerkhoff and Schenk, 2012*), the costs and availability of MRI have to be weighed as well as the patient's potential burden associated with lying in the narrow tube of an MRI scanner. In addition, various exclusion criteria for MRI would have to be taken into account.

Finally, we would like to emphasize that the reported MVS effects on spatial orienting and attention are likely present during any fMRI study, at least at 3T. Based on the present observation that the MVS-induced changes in VOR predicted the observed changes in visual search and SSA, it is very likely that these effects do scale linearly with magnetic field strength, as is the case for the VOR (e.g. *Boegle et al., 2016*). Future research needs to investigate this expectation. Of interest is also the question on the impact of MVS-related changes in spatial attention and orientation in scanning conditions under regular lighting conditions, in which at least the VOR can be suppressed. Yet, despite suppression of the VOR, we know that vestibular information pertains in the central nervous system (e.g. *Buettner and Büttner, 1979*; *Angelaki and Cullen, 2008*) and thus can still exert its influence on spatial orienting and attention. For instance, when vestibular nystagmus is suppressed by fixating a small spot of light, there is little change in the maximal firing rate of single neurons recorded in the vestibular nuclei of monkeys: the activity in 80 % of the neurons is reduced by only less than 10 % (*Buettner and Büttner, 1979*). In correspondence, behavioral studies performed under regular lighting conditions have documented an influence of vestibular stimulation on spatial cognition (see *Ferrè and Haggard, 2020* for a review). Finally, uncovering the impact of MVS on brain activity (with and without behavioral task) will be important as well. *Boegle et al., 2016* have pioneered fMRI-investigations of MVS-effects. So far, they could show that MVS had an impact on resting state activity in the default mode network and in particular in those areas related to vestibular and oculomotor function. In their follow-up work, these authors further demonstrated that MVS indeed did induce a modulation of visual-vestibular network activity (*Boegle et al., 2017*). Such neural consequences of MVS should – together with MVS effects on spatial orienting, overt attention, and VOR – be critically considered in any fMRI study.

## Materials and methods
### Subjects
We recruited seventeen healthy subjects for our experiment (5 males; average age 25.1 ± 5.0 years SD). All subjects reported to be right-handed, had normal or corrected to normal vision, did not report any vestibular or neurological disease, and provided their informed consent according to our institutional ethics board guidelines prior to our experiment. Sample size was guided by two separate power analyses ($\alpha$ = 0.025; power = 97.5%, each) informed by the studies by *Boegle et al., 2016* (3T MVS-induced VOR; data from their Figure 1a) and *Karnath et al., 1996* (changes of SSA due to CVS and neck muscle vibration; data from their Table 1), respectively. Both analyses suggest a minimum sample size of 13 subjects (one-tailed one-sample/paired t-test). To account for putative subject dropouts, we decided to recruit four additional subjects, amounting to a total sample size of 17 subjects.

### MRI setting
We used a 3T Siemens MAGNETOM Prisma MRI Scanner to apply MVS. No radio frequency (RF) or gradient coil fields were applied. Given standard subject positioning in neurology/neuroscience (subjects entering the scanner with their head first), the magnetic field vector of our MRI system pointed from subjects' toes to their head. Note that this is the exact opposite direction as compared to the situation in the study of *Roberts et al., 2011*. Accordingly, the MVS-induced VOR effects should be towards the right and thus in the opposite horizontal direction as compared to theirs. A map of the magnetic fieldlines of our system is provided in *Figure 1—figure supplement 1*. This figure also details the location of our subjects' head for the *outside 1 and 2* phases as well as for the *inside* phase.

We tried to maximize the effects of MVS on the horizontal canals by positioning subjects in a way that their head would tilt slightly backwards inside the head-coil (Siemens Head/Neck 20 A 3T Tim Coil). According to the results reported by *Roberts et al., 2011* and *Boegle et al., 2016*,

such 'pitching-up' of subject's head can increase MVS-effects on the VOR. Based on these authors' results we approximated a head pitch angle of about –30° (canthus-tragus-line vs. vertical) to consistently achieve strong MVS-effects in all subjects. To this end, we placed various cushions underneath subjects' neck. We also placed cushions to the left/right of subjects' head inside the head-coil to prevent any head movements.

## Stimulus generation

A WIN10 laptop PC was used to control our experiment through custom scripts using MATLAB R2015b 32bit (MathWorks) in combination with Cogent 2000 and Cogent Graphics (by FIL, ICN, LON at the Wellcome Department of Imaging Neuroscience, University College London) as well as with the MATLAB Support Package for Arduino. Light stimuli were generated through eight red LEDs (Type: L-513HD; Dropping resistor: 58 kΩ) using an Arduino UNO R3-compatible microcontroller (Funduino UNO R3) that was controlled by MATLAB. The intensity of six of those LEDs could be adjusted using the Arduino's pulse code modulated (PCM) analog voltage output (0 V-5 V). The remainder of the LEDs could only be switched on/off (0 V or 5 V). The LED-light was fed into a bundle of 8 glass-fiber cables that went from the MRI-control room into the scanner room at predefined locations within the subject's search field (fiber-diameter: 1 mm ≙ 0.05° visual angle; viewing distance = 1.1 m; note that viewing distance corresponded to the 'resting state' of eye convergence in darkness; see *Owens and Liebowitz, 1980*). By this means we were able to deliver dim light stimuli through the fiber-ends inside the scanner. Apart from these transient light stimuli, subjects remained in complete darkness.

During the visual search task the voltage of the six search target LEDs was PCM-modulated across the 5 seconds presentation time for a given target, starting with 0.1 V while doubling voltage in 1 second steps up to 1.6 V. Through this manipulation, we slowly increased the visibility of the search targets to increase the likelihood that subjects would find them. Remember that for most of the time (i.e. 140 s) no target was visible at all. The final central target, which was presented for 2 s at the end of the visual search task, (as well as all calibration targets and the initial central target of the looking straight ahead task) were always presented with full intensity (5 V). The six search targets were at the following x/y–locations (values denote visual angle; positive values represent rightward/upward directions, respectively): –6°/5°; 6°/5°; –6°/–5°; 6°/–5°; –12°/–1°; 12°/1°. Target presentations were separated by seven inter-target-intervals (ITIs), in which subjects performed their search in complete darkness (durations: 5 s, 10 s, 15 s, 20 s, 25 s, 30 s, 35 s). Importantly, in our visual search task the order of targets and ITIs were pseudorandomized across task phases and between participants. The additional visual search training in *outside phase 1* comprised of a subset of four of these targets and 5 ITIs (5 s, 10 s, 15 s, 20 s, 25 s; total task duration 72 s). The final central search target was always at 0°/0°. The same central target was shown during the initial 5 seconds of the looking straight ahead task. Finally, the following five targets served for eye-calibration: 0°,0°; –6°/5°; 6°/5°; –6°/–5°; 6°/–5°. This sequence of calibration targets was presented twice and targets were shown for 2 s each (total task duration 20 s; note that in one subject, the calibration task had to be repeated after the looking straight ahead task of *outside phase 1*).

## Time course of experiment

Average duration for the completion of all tasks (including additional breaks and instructions) amounted to 512s ± 44 s SD in the *outside 1* phase. Due to the additional search training in this phase, this duration was larger than for the *inside* phase and for the *outside 2* phase, which lasted 349s ± 15 s SD and 347s ± 12 s SD, respectively. The average inter-phase-intervals amounted to 233s ± 87 s SD between *outside 1* and *inside* phases as well as 241s ± 67 s SD between the *inside* phase and *outside 2* phase. The time of subjects' head being in the center of the MRI-scanner was about 10 min on average. We ensured that our experimental measures were only obtained approximately 3 min after subjects entered the MRI-scanner for the *inside* phase as well as 3 min after subjects left the scanner for the *outside* 2 phase. This was chiefly done to reduce putative shorter term aftereffects of any MVS-induced set-point adaptation on our experimental measures. VOR aftereffects with a time constant on the order of 2 min have been reported for prolonged exposure to 7T magnetic fields (compare *Jareonsettasin et al., 2016*). At the same time, this procedure should guarantee relatively constant levels of VOR across tasks in the *inside* phase. The time-courses of eye velocity in the middle column of *Figure 1* nicely demonstrate that already during the initial looking straight ahead task smooth eye

velocity was (apart from an initial influence of visual fixation) stable throughout the measurement in all phases. Note that these time-courses are also representative for our other subjects.

## Manual responses

During the search task, subjects were asked to find and fixate any target that they would spot. In addition, we asked them to press a button on a MRI-compatible response pad (5-Button Diamond Response Pad; Current Designs), whenever they found a search target. This instruction chiefly served to keep subjects' search motivation high throughout the search period. As mentioned above, we were not interested in target hits but in subjects' exploratory scan paths when no target was present. Thus, we did not systematically analyze subjects' responses. Still we provide an estimate of response performance for the *inside* and *outside* 2 phases, in which we collected data in the vast majority of subjects (n = 14 and n = 15 [manual responses were not reliably recorded in all subjects for technical reasons]): hit-rates were 91.7% ± 17.0% SD and 93.3% ± 13.8% SD, and average reaction times were 3236ms ± 698 ms SD and 2995ms ± 697 ms SD, respectively.

## Eye tracking

The position of subjects' right eye was monitored at 50 Hz sampling rate with an MR-compatible camera with integrated infrared LED illumination (MRC Systems; Model: 12M-i IR-LED). We used the ViewPoint Monocular Integrator System and the View Point Software (Arrington Research, software version 2.8.3.437) to digitize the eye-camera video and to obtain uncalibrated eye-position data (by means of dark pupil tracking). Eye-tracking was realized on a different WIN PC that was remote-controlled through the ViewPoint Ethernet-Client running on our laptop PC. Eye movement analyses were performed off-line using custom routines written in MATLAB R2017b (MathWorks). In brief, eye position samples were filtered using a second-order 10 Hz digital low-pass filter. A five-point calibration was performed based on the data obtained in our calibration task. In addition, we compensated for any tonic eye position offset in all other tasks as well. Compensation was performed separately for each task and phase, namely by removing the (average) difference in position between the visible fixation/search target(s) and eye position during target fixation(s) from the eye position record. Eye velocity was calculated based on two-point differentiation of our eye position data. Saccades were detected using an absolute eye velocity threshold of >15°/s. Saccade-onset was defined as the sample prior threshold-crossing. Saccade-offset was defined accordingly, namely as the first sample after eye velocity dropped below the threshold (compare corresponding saccade endpoints depicted as cyan circles in *Figure 1*). Time periods with blink artifacts were excluded from analyses. To obtain our velocity estimates for MVS-induced VOR we used de-saccaded horizontal/vertical eye-velocity traces (time-periods from saccade-onset to saccade-offset were treated as missing values; compare middle column of *Figure 1*).

## Statistical analyses

Apart from the de-saccaded eye velocity estimates obtained at the *inside* phase, all other data of interest (and their respective paired differences between phases) were normally distributed (Shapiro-Wilk-Test; p ≥ 0.01). Accordingly, we applied paired t-tests (alpha = 5%) to these latter data, namely between *inside* and *outside 1*, between *inside* and *outside 2*, and between *outside 1* and 2 phases, respectively. For statistical comparison of the de-saccaded eye velocity estimates during the *inside* phase with both *outside 1* and 2 phases, we log-transformed the respective paired differences to ensure normal distribution (Shapiro-Wilk-Test; p ≥ 0.01). These log-transformed data were further analyzed by means of one sample t-tests (alpha = 5%). Due to clear directional hypotheses concerning MVS-effects on behavior between *inside* and *outside 1* and 2 phases we applied one-tailed tests. Two-tailed tests were applied when comparing *outside 1* and 2 phases and when analyzing vertical eye velocity, which should be largely unaffected by MVS. Note, that for our main experiment all reported p-values also survived Bonferroni-correction for multiple testing within each measure of interest (adjusted alpha = 1.7%). Effect size estimates (Hedges $g_1 \pm CI_{95\%}$) were calculated using the Matlab Toolbox 'Measures of Effect Size' (Version 1.6; by H. Hentschke and M.C. Stüttgen).

## Acknowledgements

This work was supported by the Deutsche Forschungsgemeinschaft (KA 1258/23–1). Daniel Wiesen was supported by the Luxembourg National Research Fund (FNR/11601161). We acknowledge support by the Open Access Publishing Fund of the University of Tübingen. We particularly thank Hannah Rosenzopf and Stefan Smaczny for assisting us during our measurements. We also thank them, Marc Himmelbach and all reviewers for their suggestions as well as Michael Erb for general MRI-support.

# Additional information

## Funding

| Funder | Grant reference number | Author |
| --- | --- | --- |
| Deutsche Forschungsgemeinschaft | KA 1258/23-1 | Hans-Otto Karnath |
| Luxembourg National Research Fund | FNR/11601161 | Daniel Wiesen |
| Open Access Publishing Fund, University of Tübingen | | Axel Lindner Daniel Wiesen Hans-Otto Karnath |

The funders had no role in study design, data collection and interpretation, or the decision to submit the work for publication.

## Author contributions

Axel Lindner, Conceptualization, Formal analysis, Investigation, Methodology, Software, Visualization, Writing – original draft, Writing – review and editing; Daniel Wiesen, Conceptualization, Funding acquisition, Investigation, Methodology, Writing – review and editing; Hans-Otto Karnath, Conceptualization, Funding acquisition, Investigation, Methodology, Supervision, Writing – original draft, Writing – review and editing

## Author ORCIDs

Axel Lindner http://orcid.org/0000-0002-8201-788X
Daniel Wiesen http://orcid.org/0000-0003-3805-6627
Hans-Otto Karnath http://orcid.org/0000-0002-5518-405X

## Ethics

Human subjects: All subjects provided their informed consent and consent to publish prior to our experiment. Our experiments were approved by the ethics board of the medical faculty at the University of Tübingen (811/2016BO1).

## Decision letter and Author response

Decision letter https://doi.org/10.7554/71076.sa1
Author response https://doi.org/10.7554/71076.sa2

# Additional files

## Supplementary files

• Transparent reporting form

## Data availability

All data and custom code are available at Dryad.

The following dataset was generated:

| Author(s) | Year | Dataset title | Dataset URL | Database and Identifier |
|---|---|---|---|---|
| Lindner A, Wiesen D, Karnath H-O | 2021 | Lying in a 3T MRI Scanner Induces Neglect-Like Spatial Attention Bias | https://doi.org/10.5061/dryad.6t1g1jx05 | Dryad Digital Repository, 10.5061/dryad.6t1g1jx05 |

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
