## [Decision Letter]

**Acceptance summary:**

This paper examines the visual-ocular response in participants when exposed to the static magnetic field of a 3T MRI system. Historically, this problem has been approached from a safety perspective. In the present study, the authors ask about the behavioral consequences of this field given that it induces a response in the vestibular system, hypothesized to mimic that of a caloric vestibular stimulation event. As such, one should anticipate a biased vestibulo-ocular reflex in the static field as well as biases in spatial attention. These predictions were confirmed, with the attentional bias manifest in eye movements during a visual search task. This is an important finding because it reveals functional "artifacts" that may arise during fMRI studies, effects that may need to be considered by those conducting research in the MR environment (especially functional studies).

**Decision letter after peer review:**

Thank you for submitting your article "Lying in a 3T MRI Scanner Induces Neglect-Like Spatial Attention Bias" for consideration by *eLife*. Your article has been reviewed by 3 peer reviewers, and the evaluation has been overseen by Richard Ivry as Reviewing Editor and Senior Editor. The following individuals involved in review of your submission have agreed to reveal their identity: Bryan Ward (Reviewer #2); David Wilkinson (Reviewer #3).

The reviewers have discussed their reviews with one another and the Reviewing Editor and all had a favorable impression of the study. The results are solid and reveal an unappreciated bias that arises in the MR environment, especially at higher fields, one that will be of interest to the MR community. We have some comments for revision, although these are mostly on the clarification side.

Essential revisions:

1. The relationship between the VOR and the visual tasks was not analyzed, a missed opportunity of a potentially key relationship. A process of adaptation occurs in the MRI machine that is reflected in the after effect where nystagmus (and sense of rotation) reverses direction if the participant spends more than a few minutes in the MRI. These effects occur on a time scale that would be lost by averaging over a 10-minute period.

2. Studies have found that the perception of vertigo is shorter than the nystagmus time-constant. What is the strength of the relationship between VOR slow-phase eye velocity and saccade bias? As commented on in the supplement material, typical MVS VOR shows adaptation within a minute or two. Does the VOR and saccade bias follow a similar time-constant?

3. Subjects spent on average 10 minutes in the 3T MRI scanner (Figure S2). Studies have shown an aftereffect that occurs after leaving the magnet, the duration of which is proportional to the amount of adaptation that occurs in the scanner (Jareonsetasin et al.,). This after effect is not present for short duration exposure (Roberts 2011). The comment on line 140 directing to the supplemental video to support a lack of after effect could be misleading. The supplemental video was a short duration in the scanner and there was no appreciable aftereffect, however, there ought to be one in some of the participants after longer duration as noted in the supplemental discussion. This would most affect the looking straight ahead measure, since this was assessed first, and would occur during the peak of eye velocity both immediately after entering the magnet and immediately after exiting the magnet during the aftereffect.

4. The manuscript would be strengthened by a comment on the generalizability of the findings given the relatively restricted viewing conditions employed in the experiment. The tasks (visual search and 'straight-ahead') were conducted in complete darkness which is an uncommon viewing condition and one that would likely produce the largest bias. Would this effect be measurable under conditions in which other stimuli are being viewed? Similarly, would the effects be measurable in lower fields (e.g., 1.5T system)?

5. The discussion of how MRI scanners may be useful to help manage symptoms of neglect is a bit of a stretch here and if retained should be expanded to note the problems/challenges that one might anticipate. One questions the effectiveness of this approach given (1) the cost and availability of MRI (say relative to the galvanic and caloric vestibular stimulators that are available and which enable a more diverse range of stimulation protocols), and (2) the patient burden associated with having to lie inside a high-strength scanner, likely exacerbated in the case of neglect by the presence of allied motor impairment and/or contraindicated by the presence of metal inside the body.

---

## [Author Response]

Essential revisions:1. The relationship between the VOR and the visual tasks was not analyzed, a missed opportunity of a potentially key relationship. A process of adaptation occurs in the MRI machine that is reflected in the after effect where nystagmus (and sense of rotation) reverses direction if the participant spends more than a few minutes in the MRI. These effects occur on a time scale that would be lost by averaging over a 10-minute period.

We address this point in our reply to point 2.

2. Studies have found that the perception of vertigo is shorter than the nystagmus time-constant. What is the strength of the relationship between VOR slow-phase eye velocity and saccade bias? As commented on in the supplement material, typical MVS VOR shows adaptation within a minute or two. Does the VOR and saccade bias follow a similar time-constant?

We now provide a correlation between MVS-induced changes in VOR and respective changes in SSA and visual search (see lines 242-246 and Figure 2—figure supplement 2). Perhaps the VOR can serve as the most direct empirical proxy for the size of MVS-effects and it is informative to relate MVS-induced changes in VOR to our other experimental measures. Indeed, the changes in VOR reliably predict both the change in SSA (accounting for 46% of variance) and in the horizontal mean of visual search (explaining 53% of variance). Yet, the interdependency between these measures is anything else but straightforward as varying tasks at hand themselves have strong impact on VOR gain (e.g. compare Barr CC, Schultheis LW, Robinson DA. 1976. Voluntary, non-visual control of the human vestibulo-ocular reflex. Acta Otolaryngol 81:365-75).

We agree that it would be interesting to look at the time-course of the reported spatial bias and to also probe for MVS-adaptation effects that have been reported for the VOR. However, our study was not designed to quantify the time-course of such effects or to probe their adaptation. This is an important question for future studies, as we now also point out in our discussion. Instead, we here optimized our paradigm to reveal whether or not a qualitative influence of MVS on overt spatial attention and the SSA existed. To this end we kept subjects time in the scanner as short as possible. On the other hand, we ensured that our main measures (looking straight ahead and visual search task) only start approximately 3 minutes after subjects had entered the scanner (inside phase) and after they left the scanner (outside 2 phase). This should guarantee relatively constant levels of MVS and should help to reduce MVS aftereffects (as INDIRECTLY inferred from VOR measures; Jareonsettasin P, Otero-Millan J, Ward BK, Roberts DC, Schubert MC, Zee DS. 2016. Multiple Time Courses of Vestibular Set-Point Adaptation Revealed by Sustained Magnetic Field Stimulation of the Labyrinth. Curr Biol 26: 1359-66). We highlight these aspects now in the Discussion section (lines 271-282) and also provide further details about this rationale of our experimental design in the methods section (lines 414-429).

3. Subjects spent on average 10 minutes in the 3T MRI scanner (Figure S2). Studies have shown an aftereffect that occurs after leaving the magnet, the duration of which is proportional to the amount of adaptation that occurs in the scanner (Jareonsetasin et al.,). This after effect is not present for short duration exposure (Roberts 2011). The comment on line 140 directing to the supplemental video to support a lack of after effect could be misleading. The supplemental video was a short duration in the scanner and there was no appreciable aftereffect, however, there ought to be one in some of the participants after longer duration as noted in the supplemental discussion. This would most affect the looking straight ahead measure, since this was assessed first, and would occur during the peak of eye velocity both immediately after entering the magnet and immediately after exiting the magnet during the aftereffect.

We removed the potentially misleading reference to the supplemental video (now referred to as Video 1) and moved it to a more appropriate location (compare line 144). In addition we explicitly mention in the legend of Video 1 that prolonged exposure to the static magnetic field would yield a VOR-aftereffect that is not visible for short exposure times (lines 690-698). We also refer to the work of Roberts et al., (2011) who have nicely demonstrated this difference in their initial description of the MVS-effects.

As we already pointed out in our reply to point 2, we tried to minimize potential MVS after-effects in our study by waiting approximately for 3 minutes before continuing our measurements after subjects have left the scanner (compare methods, lines 414-429). Still, the order of experiments might play a role and the influence of MVS in the inside phase and of MVS-after effects in the outside 2 phase might have a larger impact on the SSA than on visual search. As we point out above (see our response addressing point 2), it will be interesting to look at the time-course von MVS-induced spatial biases in future studies. Our experiment does not allow to address this issue but merely intended to investigate whether or not there is a qualitative influence of MVS on spatial attention and exploration. We also acknowledge this aspect now in our paper (lines 271-282).

4. The manuscript would be strengthened by a comment on the generalizability of the findings given the relatively restricted viewing conditions employed in the experiment. The tasks (visual search and 'straight-ahead') were conducted in complete darkness which is an uncommon viewing condition and one that would likely produce the largest bias. Would this effect be measurable under conditions in which other stimuli are being viewed? Similarly, would the effects be measurable in lower fields (e.g., 1.5T system)?

These are important aspects, which we are happy to add to our discussion. We now speculate about the presence of the effects under more “naturalistic” viewing conditions (see lines 336-344):

“Of interest is also the question on the impact of MVS-related changes in spatial attention and orientation in scanning conditions under regular lighting conditions, in which at least the VOR can be suppressed. Yet, despite suppression of the VOR, we know that vestibular information pertains in the central nervous system (e.g. Buettner and Büttner 1979; Angelaki and Cullen 2008) and thus can still exert its influence on spatial orienting and attention. For instance, when vestibular nystagmus is suppressed by fixating a small spot of light, there is little change in the maximal firing rate of single neurons recorded in the vestibular nuclei of monkeys: the activity in 80% of the neurons is reduced by only less than 10% (Buettner and Büttner 1979). In correspondence, behavioral studies performed under regular lighting conditions have documented an influence of vestibular stimulation on spatial cognition (see Ferrè and Haggard 2020 for a review).”

We also raise the question about how field strength would modulate these effects (see lines 333-336): “Based on the present observation that the MVS-induced changes in VOR predicted the observed changes in visual search and SSA, it is very likely that these effects do scale linearly with magnetic field strength, as is the case for the VOR (e.g. Boegle et al., 2016). Future research needs to investigate this expectation”.

5. The discussion of how MRI scanners may be useful to help manage symptoms of neglect is a bit of a stretch here and if retained should be expanded to note the problems/challenges that one might anticipate. One questions the effectiveness of this approach given (1) the cost and availability of MRI (say relative to the galvanic and caloric vestibular stimulators that are available and which enable a more diverse range of stimulation protocols), and (2) the patient burden associated with having to lie inside a high-strength scanner, likely exacerbated in the case of neglect by the presence of allied motor impairment and/or contraindicated by the presence of metal inside the body.

We agree with the referee and included his/her arguments; we now added (lines 323-327):

“Against this possible advantage over CVS and over other types of vestibular stimulation (Kerkhoff and Schenk 2012), the costs and availability of MRI have to be weighed as well as the patient’s potential burden associated with lying in the narrow tube of an MRI scanner. In addition, various exclusion criteria for MRI would have to be taken into account”.